# Genetic and Structural Diversity of Prokaryotic Ice-Binding Proteins from the Central Arctic Ocean

**DOI:** 10.3390/genes14020363

**Published:** 2023-01-30

**Authors:** Johanna C. Winder, William Boulton, Asaf Salamov, Sarah Lena Eggers, Katja Metfies, Vincent Moulton, Thomas Mock

**Affiliations:** 1School of Environmental Sciences, University of East Anglia, Norwich Research Park, Norwich NR4 7TJ, UK; 2School of Computing Sciences, University of East Anglia, Norwich Research Park, Norwich NR4 7TJ, UK; 3DOE Joint Genome Institute, Algal and Fungal Program, Lawrence Berkeley National Laboratory, 1 Cyclotron Road, Berkeley, CA 94720, USA; 4Alfred Wegener Institute, Polar Biological Oceanography, Am Handelshafen 12, 27570 Bremerhaven, Germany

**Keywords:** metagenomics, MAGs, ice-binding proteins, DUF3494, domain shuffling, polar genomics, Arctic Ocean, MOSAiC expedition

## Abstract

Ice-binding proteins (IBPs) are a group of ecologically and biotechnologically relevant enzymes produced by psychrophilic organisms. Although putative IBPs containing the domain of unknown function (DUF) 3494 have been identified in many taxa of polar microbes, our knowledge of their genetic and structural diversity in natural microbial communities is limited. Here, we used samples from sea ice and sea water collected in the central Arctic Ocean as part of the MOSAiC expedition for metagenome sequencing and the subsequent analyses of metagenome-assembled genomes (MAGs). By linking structurally diverse IBPs to particular environments and potential functions, we reveal that IBP sequences are enriched in interior ice, have diverse genomic contexts and cluster taxonomically. Their diverse protein structures may be a consequence of domain shuffling, leading to variable combinations of protein domains in IBPs and probably reflecting the functional versatility required to thrive in the extreme and variable environment of the central Arctic Ocean.

## 1. Introduction

Ice-binding proteins (IBPs) are a large group of cold-active enzymes found across all three domains of life, but little is known about their diversity in natural environments. Depending on their concentration, IBPs function in one of two dominant modes: thermal hysteresis (TH) or ice-recrystallisation inhibition (IRI) [1]. TH refers to freezing point depression, while IRI prevents the growth of larger, tissue-damaging ice crystals [2,3]. Which of these modes dominates is also thought to relate to their environmental function [1]. In prokaryotic and eukaryotic microbes, the majority of ice-binding proteins contain a ~200 amino acid domain of unknown function 3494 (DUF 3494) [4]. These DUF3494 IBPs (henceforth IBPs) are often found in psychrophilic bacteria [5], in part due to prevalent horizontal gene transfer (HGT) [4,6]. Our understanding of the function of prokaryotic IBPs is mainly derived from lab-based studies, but how widespread or representative these functions are remains unknown.

A number of bacterial IBPs have been functionally characterised, revealing varied potential environmental roles related to their structures. The Pfam library reports over 4000 IBP sequences from over 3000 taxa, the majority of which are prokaryotic [5]. Among them, 237 domain architectures are found [5]. Despite this diversity, studies of prokaryotic IBPs have largely focused on targeted, lab-based studies of single IBPs. The majority of characterised IBPs have a single domain architecture with an N-terminal signal peptide, implying secretion or membrane localisation [4,5]. Roles have been suggested for different prokaryotic IBP structures—including the prevention of heterogeneous ice formation by the organism [7,8] and the maintenance of a liquid habitat by conserving triple junctions between ice grains [9,10]. IBPs from *Shewanella frigidimarina* and *Marinomonas primoryensis* contain bacterial immunoglobulin-like repeats which act as a tether between the cell membrane and the IBP, permitting an ice adhesion function [11,12]. Ig-like domains generally consist of two antiparallel β-sheets which twist to surround a hydrophobic core, and are often associated with bacterial adhesion to a variety of substrates [13]. In addition to these diverse functions of prokaryotic IBPs, a number of microbial eukaryotic IBPs have been functionally characterised, suggesting roles in brine channel shaping [14] and intracellular roles [15].

The diversity of IBPs may be connected to their environmental, taxonomic and specific genomic contexts. Prokaryotic IBPs are found in various frozen environments, including glacier cryoconites [16], subglacial lakes [9], polar desert soils [17] and sea ice [7]. However, the study of the natural diversity of IBPs in these environments is limited by their accessibility. This is especially true in winter, when IBPs may play an important role [18]. Metagenomics is a powerful tool to explore both the taxonomic and functional compositions of microbial communities, as well as to explore a specific group of sequences such as IBPs. Due to the limited accessibility of polar environments, only a few meta-omics studies have addressed IBPs directly or indirectly, but this has still provided insight into their ecology. These studies have revealed eukaryotic IBPs to be highly expressed in situ [19] and implicated prokaryotic IBPs in a commensal relationship with an Antarctic moss [20]. Some metagenomics studies have also referenced IBPs in passing, but without focusing further [17,21]. Metagenomics can yield the sequences of abundantly encoded IBPs, providing information about their taxonomic distribution and allowing the prediction of and comparison between their sequences and structures. This can then be used to suggest how diversity is generated within a taxon, as, for example, by domain shuffling. Gene synteny is also especially relevant in bacteria, where neighbouring genes are mostly co-transcribed in operons [22], and horizontally transferred genes can cluster in chromosomal “hotspots” [23]. Metagenome-assembled genomes (MAGs) can be used to address this topic, while avoiding the complications of culture-based approaches [24,25]. These have rarely been applied to polar contexts, despite the challenges of culturing organisms from these environments [26,27,28,29].

Here, we used metagenome-informed genomics to explore the genetic diversity and predicted structural diversity of prokaryotic DUF3494 IBPs from the central Arctic Ocean. During leg 2 of the MOSAiC expedition, 15 metagenomic samples were collected spanning the bathypelagic, mesopelagic, epipelagic, sea–ice interface and interior ice layers [30]. Unlike in many other studies, these samples were collected during polar winter. We explored the total community composition as well as the composition of IBP-encoding taxa within each environment, expecting that the metagenomes from the sea–ice interface and the interior ice would have a higher relative abundance of IBP genes compared to those from epipelagic or meso/bathypelagic environments. We characterised the diverse possible IBP domain architectures present in the samples, predicting the structures of the abundant architectures. MAGs were used to explore the genomic context of IBPs, determining which domain architectures were abundant in the genes upstream and downstream of the IBPs. We then compared the amino acid sequences of DUF3494s found in IBPs with abundant domain architectures to determine structural or taxonomic trends. Finally we compared the amino acid sequences of every DUF3494, exploring the distribution of domain architectures, signal peptide presence and transmembrane domain presence.

## 2. Materials and Methods

### 2.1. Sample Collection

Fifteen metagenome samples were collected during leg 2 of the MOSAiC expedition (collection dates between 13 January 2019 and 7 February 2020), during the Arctic winter (Figure 1). These samples were collected both from pelagic layers, with seawater collected via sampling from a CTD rosette, and from sea–ice layers. Ice samples were melted, and 50 mL of sterile filtered seatwater were added per 1 cm ice core. Samples were filtered with a Sterivex 0.22 micrometre filter, stored at −80 °C on board the Polarstern until the end of leg 2 (24 February 2020), and subsequently shipped to the Alfred Wegener Institute, at a temperature of −80 °C. The sample volumes used can be found in Appendix A. Two of the fifteen samples were created through pooling; i.e., the third in each trio of epipelagic samples was pooled from the other two (pelagic samples from the same CTD rosette). Together, these 15 metagenomic samples constituted the set of ECO-omics metagenome pilot samples. Of the 15 samples, 8 were from pelagic layers and the remaining 7 from sea-ice. Of the seawater samples, 4 were from the epipelagic, with 2 taken from a depth of 20 m and 2 from 50 m, and a further 2 samples were generated through pooling material from the other 2 replicates (see Appendix A for details). Each pair was collected from the same CTD rosette. The remaining two seawater samples were from the meso and bathypelagic, sampled from depths of 200 and 4082 m, respectively. Of the seven sea-ice samples, five co-located samples, including the four samples labelled interior ice, were from different layers within the same ice core, from first-year ice. The remaining two samples were second-year ice from the sea–ice interface, the 0 to 5 cm bottom layer of the ice, at the interface with the ocean. Associated metadata are in Appendix A, which also provide the IDs of the relevant GOLD databases.

### 2.2. DNA Extraction and Sequencing

The DNA was extracted at the Alfred Wegener Institute, using the Qiagen PowerWater DNA kit, following a slightly modified version of the QIAGEN DNeasy Power Water SOP v1 (QIAGEN N.V., Hilden, Germany) [31]. Samples were sent to the DoE Joint Genome Institute (JGI) for sequencing. Sequencing was performed following either the Illumina regular fragment, 300 base pair, or the Illumina low input, 300 base pair protocols (Appendix A), with the sea–ice interface and meso and bathypelagic samples following the low input protocol, and epipelagic and interior ice samples using the regular fragment protocol.

For the regular protocol, the DNA was sheared to 300 bp using the Covaris LE220-Plus and size selected with SPRI using TotalPure NGS beads (Omega Bio-tek, Norcross, GA, USA). The fragments were treated with end-repair, A-tailing and the ligation of Illumina compatible adapters (IDT, Inc, Gladesville, Australia) using the KAPA-HyperPrep kit (KAPA Biosystems, Wilmington, MA, USA). The prepared libraries were quantified using KAPA Biosystems’ next-generation sequencing library qPCR kit and run on a Roche LightCycler 480 real-time PCR instrument. The sequencing of the flowcell was performed with the Illumina NovaSeq sequencer using NovaSeq XP V1.5 reagent kits, S4 flowcell, following a 2 × 151 indexed run recipe. For the low input protocol (10 ng of DNA), the procedure was the same, except that the sample was enriched using 5 cycles of PCR.

#### Bioinformatics Processing of Samples

The sequence quality control, assembly, annotation and binning were all performed using the IMG/M metagenome annotation pipeline (v.5.0.23) [32]. Briefly, reads were trimmed of Illumina adapters and then filtered for quality and for human or lab contamination using BBDuk (v38.79) [33], and each sample was individually assembled using metaSPAdes (v3.14.1) [34]. Genes were predicted using consensus between GeneMark (v1.05) [35], INFERNAL (v1.1.3) [36], Prodigal (v2.6.3) [37] and tRNAscan-SE (v.2.0.7) [38]. Only contigs of lengths of at least 500 base pairs were retained, representing between 61.2% and 94.7% of the total reads of the samples (Appendix A). Annotations were performed using the hmmsearch function of HMMER (3.1b2) [39], using model specific cutoffs, and with models from a range of protein databases including Pfam-A (v30) [40]. Bins were generated using metaBAT2 (v2.12.1) [41], with a minimum contig size of 1000, and assessed for completeness and contamination with CheckM (v1.0.12) [42], and bins of less than 50% completeness or above 10% contamination were discarded. The bins were taxonomically assigned using GTDB-tk (v0.2.2) [43]. The subsequent analysis of the IBPs used all genes that were annotated with the PF11999 Pfam domain. The abundance of the PF11999 was measured using reads per kilobase million (RPKM). We used the Phobius web server [44] to further annotate transmembrane domains and signal peptides, and MMSeqs2 (v01889*) [45] to scan the assemblies against both the NR and MMETSP [46] databases for taxonomic annotation. Sequences classified as eukaryotic were removed for downstream analysis.

### 2.3. Community Analysis

We compared the prokaryotic community compositions of the total assembly, the MAGs, and their respective IBP-producing communities across sites. We used the R packages phyloseq (v1.40.0) and ggplot2 (v3.4.0) [47,48] to plot both the total prokaryotic community composition and that of the IBP-containing community. Vegan in R (v2.6-4) [49] was used to carry out the comparisons of community composition, using permANOVA and non-metric multidimensional scaling (NMDS) to visualise them.

We then explored which bacterial orders encoded IBPs with diverse gene architectures—defined as containing >1 domain in the IBP or containing a signal peptide and/or transmembrane domain(s).

### 2.4. Protein Structure Prediction

The domain architectures for modelling were identified by the presence of multiple protein families (Pfams) within the same gene. We selected the five most environmentally abundant (total reads per kilobase million; RPKM) domain architectures in the total dataset for modelling. Representative IBPs for each domain architecture were further selected on the basis of their environmental abundance. The structures were modelled using AlphaFold (v2.1.1) [50], with the models reported being the highest confidence models from the AlphaFold output. Functional information about the individual domains in these IBPs was obtained from the Interpro database [5]. A conceptual figure denoting typical domain architecture was produced using Inkscape (v1.2.1).

### 2.5. Upstream and Downstream Gene Analysis

The domain architecture of the genes surrounding the IBPs in MAGs was determined by querying the genes found the closest, upstream or downstream, to the IBP genes, and recording their relative locations and which protein families were present. We queried which domains and domain architectures were the most abundant within these upstream and downstream genes. Genomic context and domain architecture figures were produced using Inkscape v1.2.1. As above, broader functional characterisations were obtained using InterPro [5].

### 2.6. Phylogenetic Analysis of IBPs

To determine how the phylogenetic relationships between IBPs varied depending on the domain architecture, environment and taxonomic assignments, we produced gene trees of the most environmentally abundant gene architectures, as well as gene trees of IBPs across all domain architectures. The alignments of the amino acid sequences of HMMER hits to the DUF3494 domain were produced using muscle (v2.0.4) [51], and low quality columns of the alignment were removed using TrimAl (v1.2) [52]. The trees were generated with FastTree (v2.1.1) [53], using the default parameters, and visualised using interactive tree of life (IToL; v6.6) [54]. We repeated this method for IBPs within MAGs. Gene trees with fewer than 60 leaves, or with multi-copy DUF3494 domain architectures, were rooted at their midpoint. For the remaining trees, we rooted the trees using an outgroup of 130 IBPs from the dinoflagellate *Polarella glacialis* [55] (accessions in Appendix A).

## 3. Results

### 3.1. Diverse Prokaryotic Communities and MAGs Encode IBP Genes

From the whole metagenome assemblies, we retrieved between 4.91 × 10^7^ and 2.50 × 10^8^ bacterial reads per sample and between 1.59 × 10^5^ and 9.00 × 10^6^ archaeal reads per sample. Of all of the assemblies, 71% could be classified to the order level. From them, we identified 207 bacterial orders and 32 archaeal orders. The most commonly identified bacterial orders were Cellvibrionales (15.2%) and Rhodobacterales (13.3%). The most common archaeal orders were Nitrosopumilales (0.88%) and Candidatus Poseidoniales (0.46%). We also retrieved 750 total medium and high quality MAGs from these samples (Figure 2c,d).

The subset of these communities in which DUF3494-containing proteins (henceforth IBPs) were found was analysed separately. In all, 85.74% of IBPs could be assigned order-level taxonomy. These IBP-encoding communities were composed of 60 bacterial orders and 5 archaeal orders. The most common bacterial orders were Flavobacteriales (1936 IBPs; 50.79%) (Bacteroidetes) and Alteromonadales (893 IBPs; 23.43%) (Gammaproteobacteria) (Figure 2). The most common archaeal orders were Methanomicrobiales (0.36%; 14 IBPs) (Euryarchaeota) and Candidatus Poseidonales (0.11%; 4 IBPs) (Candidatus Thermoplasmatota). A total of 3581 (80.54%) IBPs were found in the interior ice, 797 (17.93%) in the sea–ice interface, 60 (1.35%) in the epipelagic zone and 8 (0.18%) in the meso/bathypelagic zones.

In all, 199 IBPs were encoded by 79 MAGs. The most IBP-encoding MAGs were obtained from the interior ice habitat (67/79 MAGs), followed by the sea–ice interface (8/79 MAGs), with three and one MAGs found in the epipelagic and meso/bathypelagic environments, respectively. The order level composition of the IBP-encoding communities varied among different sites (total assembly permANOVA: F = 6.83, *p* = 0.001, R2 = 0.651; MAGs (genus level): F = 4.81, *p* = 0.002, R2 = 0.74) (Appendix A).

### 3.2. Diverse IBP Structures Are Abundant in the Natural Environment

Diverse domain architectures were predicted from the genomic sequences of the IBPs. A total of 116 unique domain architectures were found in 3869 prokaryotic IBPs spanning 65 identified orders. These diverse architectures included a total of 46 protein families. Single domain IBPs were by far the most abundant in the environment (61.54% of the total environmental relative abundance, RPKM) and the most prevalent across the samples (accounting for 70.53% of the total number of IBPs), followed by double domain IBPs (20.51% of the RPKM; 11.75% of the total number of IBPs) (Figure 3c; Table 1). Triple domain IBPs were also abundant (1.15%; 0.44%) (Figure 3d and Table 1).

Some differences in the structure of the DUF3494 domain were observed. All modelled proteins contained the discontinuous right-handed β-solenoid with three flat faces and a braced α helix. However, the length of the β-solenoid varied, with longer solenoids containing 14 coils found in some of the most environmentally abundant IBPs (Figure 3e–g).

In the natural environment, IBPs containing a protein family classified as immunoglobulin-like make up a large proportion of the environmental relative abundance (467.69 RPKM; 6.62%), but they were not as prevalent across samples, accounting for only 3.57% of all of the IBPs found. They were, therefore, likely found in highly abundant individual IBPs rather than in a large number of distinct IBPs with the same architectures. Of these proteins, 15.22% contained a transmembrane domain, 45.65% contained a signal peptide and 7.25% contained both. In all, 84.06% of these IBPs came from interior ice, 14.49% from the sea–ice interface and 1.45% from the epipelagic zone.

The most prevalent domain architectures across the samples consisted of protein families whose role involves cell adhesion and exopolysaccharides. IBPs with these domain architectures had an abundance of 411.20 RPKM (5.82% of the environmental relative abundance), constituting 4.88% of all IBPs found. This is reflected in certain architectures with large numbers of repeated domains. The most striking examples of them among our samples are double domain IBPs with up to 26 C-terminal thrombospondin type-3 repeats, and single and double domain IBPs containing up to 7 C- or N-terminal bacterial immunoglobulin-like (BIg) domains. A total of 33.93% of IBPs with an adhesion function contained a TMD, while 48.68% contained a signal peptide and 32.8% contained both. As for their origins, 84.66% of these IBPs came from interior ice, 12.70% from the sea–ice interface and 2.65% from the epipelagic zone.

Other broadly abundant functions or protein families found in these domain architectures include protein families with no known or suggested functions (461.2 RPKM/6.53% abundance; 3.15% prevalence across samples), calcium binding proteins (54.88 RPKM/0.78%; 1.16%) and trafficking/secretion-related proteins (81.25 RPKM/1.15%; 0.70%).

In the MAGs, the most abundant domain architectures were single domain (55.32%; 56.28%), double domain (23.57%; 20.60%), single domain with a DUF4842 (7.86%; 4.52%), triple domain (1.64%; 1.51%) and single domain with a PEP C-term motif (1.58%; 3.02%).

### 3.3. The Genomic Context of IBPs Suggests Mechanisms for Generating Diversity

MAGs were used to explore the protein families present in the genes flanking IBP genes. Forty-six of the seventy-nine MAGs contained >1 IBP. The highest number of IBPs in a single MAG was nine (e.g., Figure 4b). In MAGs with multiple IBPs, these IBPs were frequently found in the same contig, immediately upstream or downstream of one another. Furthermore, these IBPs often had identical domain architectures, e.g., double domains (Figure 4a,d).

One hundred and three unique domain architectures were found downstream of IBPs (Figure 4). The most frequent domain architectures found downstream of IBPs in MAGs were single domain IBPs (6.12%) and double domain IBPs (4.76%). Following this, the five most abundant downstream domain architectures contained small solute membrane transport proteins (MFS; pfam07690; 3.40%), bacterial 2-component systems containing a DNA binding domain and a response regulator receiver domain (pfam04397_pfam00072; 2.72%), an antioxidant enzyme (AhpC/TSA family; pfam00578; 2.04%), DNA topoisomerase (pfam01131_pfam01751; 2.04%) and a phosphodiesterase (pfam01663; 2.04%).

The most frequent individual domains found downstream of IBPs, rather than whole architectures, were generally characterised by IBPs and repeats. They are as follows: IBPs (pfam11999; 10.82%), FG-GAP repeats (pfam14312; 3.03%), response regulator receiver domains (pfam00072; 2.60%), DNA binding domains (pfam04397; 2.60%) and bacterial transferase hexapeptides (pfam00132; 2.16%).

One hundred and seven unique domain architectures were found upstream of IBPs (Figure 4). The most frequent domain architectures found upstream of IBPs in MAGs were single domain IBPs (5.19%) and double domain IBPs (5.19%). Following this, the five most abundant upstream domain architectures were transposases (IS66 family; pfam03050; 2.60%), aminotransferases (pfam00155; 1.95%), autoregulatory aminotransferases (pfam00155_pfam00392; 1.95%), post-translational sulfatase modification domains (SUMF1; pfam03781; 1.95%), and DUF3050 (pfam11251; 1.95%).

The most frequent individual domains found upstream of IBPs were IBPs (pfam11999; 10.83%), a metal-binding motif (pfam11617; 8.33%), a β-propellor repeat (pfam07676; 4.17%), a WG repeat motif (pfam14903; 3.33%) and an aminotransferase (pfam00155; 2.50%).

### 3.4. Phylogenetic Distribution of Abundant Domain Architectures Implicates Domain Shuffling

The sequences of the DUF3494 domain(s) of IBPs with the most abundant domain architectures were compared (Figure 5 and Table 1). A number of these most abundant domain architectures did not appear to be present in a wide variety of individual IBPs—rather, individual IBPs with these architectures were highly abundant.

In all, 2886 single domain IBPs were found. Of these, just 39.92% contained a signal peptide, and 90.6% contained no transmembrane domain, while 8.34% contained one TMD, 0.90% contained two TMDs and 0.07% contained three TMDs. A total of 2.94% contained both an SP and at least one TMD. Of the 2886 total IBPs, 2501 could be classified to the order level. Among them, the five most abundant orders encoding this domain architecture were Flavobacteriales (1130 IBPs), Alteromonadales (714 IBPs), Burkholderiales (112 IBPs), Oceanospirillales (71 IBPs) and Acidimicrobiales (47 IBPs). As for their origin, 78.55% of these IBPs came from interior ice, 19.82% from the sea–ice interface, 1.42% from the epipelagic zone and 0.021% from the meso/bathypelagic zones.

A total of 455 double domain IBPs were found (Figure 5a), 57.80% of which contained a signal peptide, implying secretion. Meanwhile, 96.04% of them contained no transmembrane domain (TMD), while 3.74% contained one TMD and 0.22% contained two TMDs. Only 0.44% contained both a TMD and an SP, while 38.68% contained neither. In all, 335 IBPs with this domain architecture could be classified to the order level. The five most abundant orders were Flavobacteriales (278 IBPs), Cytophagales (10 IBPs), Alteromonadales (8 IBPs), Acidimicrobiales (6 IBPs), Burkholderiales, Cellvibrionales, Solirubrobacterales, Streptomycetales and Thiotrichales (3 each). As for their origins, 85.06% of these IBPs came from interior ice, 14.07% from the sea–ice interface and 0.88% from the epipelagic zone.

In all, 86 single domain IBPs contained a DUF4842 (pfam16130) the function of which is unknown, but which contains a β-barrel immunoglobulin fold (Figure 5b). Of that total, 37.21% contained a signal peptide, while the majority lacked one, suggesting that many of these proteins may be intracellular. Similarly, only 19.77% contained a transmembrane domain, and none contained both a signal peptide and transmembrane domain, while 43.02% contained neither. Of the 75 IBPs with this domain architecture which could be classified to the order level, 51 were found within the Alteromonadales, 17 within the Flavobacteriales, 6 within the Cellvibrionales and 1 within the Vibrionales. In total, 88.37% of these IBPs came from interior ice, 10.47% from the sea–ice interface and 1.16% from the epipelagic zone.

A total of 92 single domain IBPs contained a PEP C-term motif (pfam07589) that is exopolysaccharide-related (Figure 5c). Among them, 53.26% contained a signal peptide, implying secretion, and 61.96% contained one transmembrane domain (TMD), while 3.26% contained two TMDs. In all, 34.78% contained both a transmembrane domain and a signal peptide, while 16.30% contained neither. Of the 89 IBPs with this domain architecture which could be classified to the order level, 70 were found within Alteromonadales, 8 within Oceanospirillales, 6 within Verrucomicrobiales and 2 in Methylococcales, and the remaining 3 were found in Ferrovales, Rhodobacterales and Thiotrichales, respectively. As for their origins, 83.70% of these IBPs came from interior ice, 10.87% from the sea–ice interface and 5.43% from the epipelagic zone.

In all, 23 double domain IBPs contained a DUF11 (pfam01345), whose function is unknown but is thought to be cell-wall related. Of them, 73.91% contained a signal peptide, implying that the majority were secreted. Only 8.70% contained a transmembrane domain, and none contained both an SP and a TMD; 17.39% contained neither. Of the 22 IBPs with this domain architecture which could be classified to the order level, 20 were found within the Flavobacteriales and 2 were found within the Saprospirales. A total of 82.61% of these IBPs came from interior ice, and 17.39% from the sea–ice interface.

Seventeen triple domain IBPs were found (Figure 5d). Of these, 41.18% contained a signal peptide. Only 5.88% contained a transmembrane domain, and none contained both an SP and a TMD; 52.94% contained neither. Of the 12 IBPs with this domain architecture which could be classified to the order level, 5 were found within Flavobacteriales, 4 within Micrococcales, 2 within Cytophagales and Cellvibrionales and 1 within Thiotricales. A total of 94.12% of these IBPs came from interior ice, and 5.88% from the sea–ice interface.

### 3.5. IBPs from the Total Assembly Cluster Taxonomically

A large amount of structural diversity was distributed across the tree of 3869 IBP sequences (Figure 6). In the total assembly, 43.47% of IBPs contained a signal peptide, while 9.10% contained one TMD, 0.85% contained two TMDs, and 0.05% contained three TMDs. Only 3.23% of the IBPs contained both an SP and at least one TMD, and 49.75% contained neither.

The most abundant orders in which signal-peptide-containing IBPs were found were Flavobacteriales (49.59%), Alteromonadales (29.42%), Oceanospirillales (3.14%), Burkholderiales (2.69%) and Cytophagales (1.72%). For IBPs without signal peptides, the most abundant orders were Flavobacteriales (47.94%), Alteromonadales (25.98%), Burkholderiales (4.45%), Oceanospirillales (2.57%), and Micrococales (2.14%).

IBPs with TMDs were mostly found in the order Alteromonadales (58.58%), followed by Flavobacteriales (16.18%), Oceanospirillales (3.88%), Thiotrichales (2.27%) and Acidimicrobiales (1.94%). Those without TMDs were mostly found in Flavobacteriales (51.81%), followed by Alteromonadales (24.09%), Burkholderiales (3.94%), Oceanospirillales (2.69%) and Micrococcales (1.59%).

A total of 25.41% of IBPs had a diverse domain architecture, defined as a protein with more than one domain (IBP or other protein family) (Figure 6). Of these proteins with a diverse domain architecture, 80.26% could be classified to the order level. The five most abundant orders among them were Flavobacteriales (55.26%), Alteromonadales (21.67%), Oceanospirillales (2.41%), Cellvibrionales (2.41%) and Cytophagales (2.16%). Of the single domain IBPs (i.e., those without a diverse domain architecture), 83.75% could be classified to order level. The five most abundant orders among them were Flavobacteriales (46.13%), Alteromonadales (29.29%), Burkholderiales (4.39%), Oceanospirillales (2.94%) and Micrococcales (1.66%).

## 4. Discussion

Our findings suggest that the structural diversity of prokaryotic IBPs is associated with their taxonomy. By surveying the complement of ice-binding proteins encoded by prokaryotic sea ice and marine communities during an Arctic winter, we compared ecological and individual-scale observations. We queried environmentally abundant IBP domain architectures, linking these to broader functions as well as to genomic context and taxonomy. The IBPs were encoded by a diverse subset of communities and MAGs. IBPs containing immunoglobulin-like domains and domains involved in cell adhesion were abundant. The genomic context of the IBPs was dominated by other IBPs. The taxonomic clustering of the IBPs was sometimes also reflected in the variable presence of signal peptides and transmembrane domains. Together, these results provide new insight into the previously underexplored natural diversity of prokaryotic IBPs in the central Arctic Ocean. Furthermore, these results highlight the value of MAGs as a complement to whole metagenomes [56], especially in study regions lacking abundant reference genomes [29].

Ig-like domains and cell adhesion-related domains were the most abundant non-ice binding domains found in the IBPs. The presence of bacterial immunoglobulin (BIg) domains in IBPs has previously been attributed to an adhesin function. These domains act as a flexible tether between the IBP and the cell [11,12]. The ice-tethering function of IBPs is thought to hold bacterial cells in close proximity to the ice, where oxygen and nutrient conditions are favourable [11]. However, IBPs that play an ice-tethering role typically contain both a membrane anchor and a signal peptide [12]. The ice-binding domain extends out via a tether which is anchored in the cell membrane. Although this has been suggested as a dominant function of IBPs previously [4,12], our results provide minimal evidence for it, as only small proportions of IBPs containing Ig or Ig-like domains had both a signal peptide and a transmembrane domain. Conversely, specific adhesion-related domains (e.g., collagen triple helix repeat) were prevalent. These domains can contribute to the ability of biofilms to bind the extracellular matrix [57,58,59,60]. Sea ice harbours microbial biofilms embedded in extracellular polymeric substances [61]. These have been suggested to create microenvironments where nutrients accumulate [62]. Our results indicate a potential role for IBPs in anchoring biofilms to ice.

Single (sd) and double (dd) domain IBPs were by far the most abundant protein domain architectures; however, less than half of the sdIBPs contained a signal peptide. Although there are other pathways for secretion that are not SP-mediated [63], this suggests that a significant proportion of sdIBPs are intracellular. An intracellular IBP has been found in the plastid membrane of a sea-ice diatom [15]; however, its biological role is unknown. It is possible that these putatively intracellular IBPs function to limit the formation of intracellular ice, or play a role in the poorly-understood freezing perception [64,65]; however, this merits further exploration. Conversely, two-thirds of ddIBPs contained signal peptides, suggesting that these play an extracellular role. DUF3494 ddIBPs have been physico-chemically characterised and shown to not be inherently more active than sdIBPs [10]. Ice crystals in sea ice have variable plane orientations [66]. It is therefore possible that multidomain IBPs are preferentially secreted in order to increase the likelihood of successful adsorption to diverse ice planes.

In the MAGs, IBPs were most frequently flanked by other IBPs. Gene synteny can be used as a method of inferring the biological function of proteins [67,68,69]. Genes which cluster in prokaryotic genomes may be part of the same operon. The genes within these hypothetical operons may be connected in various ways; most notably, they may be part of the same metabolic pathway, be part of a shared non-metabolic (e.g., regulatory) pathway, or physically interact [67]. By clustering closely in the genome, IBPs, which are thought to be regulated in response to external conditions, i.e., freezing [70], could be co-regulated. Given that many IBPs are secreted and therefore function in a comparatively vast environment, they may require large volumes of protein in order to adapt rapidly to the environment [3,71]. Encoding tandem IBPs may be a mechanism to allow the bulk production of these proteins [72]. Furthermore, non-DUF3494 IBPs sometimes form multimers which function more effectively than monomeric IBPs [73,74]. Given that this is a known feature of proteins which cluster in bacterial genes, it is possible that tandem IBPs result in multimeric protein formation.

IBPs clustered taxonomically when comparing abundant domain architectures (Figure 5). There were differences between the taxonomic distributions of double domain, DUF4843-containing, PEP C-term motif-containing and triple domain IBPs. In many cases, the IBP sequences clustered according to taxonomy. Bacteria obtain IBPs via horizontal gene transfer [75]. However, our results imply that, after this acquisition, the host organisms may utilise domain shuffling to adapt the IBPs for their specific habitat and lifestyle. This is further supported by the taxonomic patterns of TMD presence. For example, transmembrane domains are abundant in PEP C-term motif-containing IBPs from Gammaproteobacteria, but clearly absent from Verrucomicrobia IBPs. PEP C-term motifs are found in biofilms and are used for protein sorting through association with exopolysaccharides in Gram negative bacteria [76]. It has been proposed that the PEP C-term motifs are necessary for bacterial aggregate formation [77]—the presence of a TMD may, therefore, alter the way that these function.

IBPs also clustered taxonomically when all sequences, not just those with abundant domain architectures, were compared (Figure 6), with IBPs from Bacteroidetes forming two distinct groups. The less basal of these groups is enriched in double domain IBPs compared to other groups, and the majority of signal peptide-containing IBPs were found in Flavobacteria (Bacteroidetes). If signal peptide addition is linked to specific taxa, this would have consequences for the ecology of IBPs, as intracellular IBPs and secreted IBPs presumably have very different roles. The observation that IBPs form taxonomic groupings has implications for their evolution, and we suggest that sdIBPs may act as building blocks for their host organisms to duplicate and shuffle into diverse architectures and, subsequently, functions.

## Figures and Tables

**Figure 1 genes-14-00363-f001:**
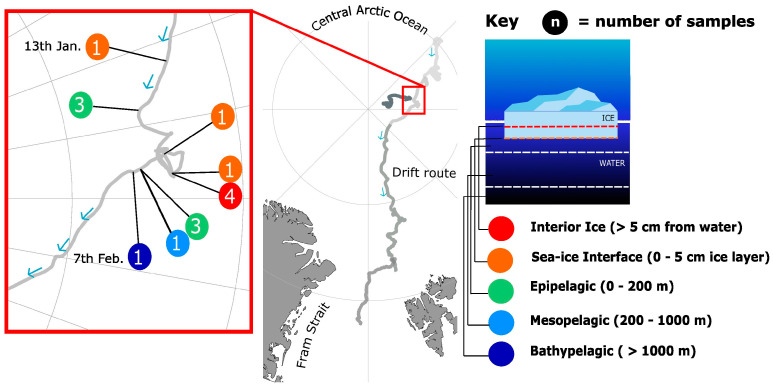
Drift route of the MOSAiC expedition, and description of the pilot samples. The red box shows the drift route of the RV Polarstern between the 13 January and the 7 February 2020. During this time, the 15 pilot samples were collected. Co-occurring samples (either from the same CTD rosette, or neighbouring ice cores) are shown, with the number of co-located samples from the same environment circled. The schematic diagram on the right describes the environment of each of the samples. Figure adapted under a CC BY 4.0 licence from [30].

**Figure 2 genes-14-00363-f002:**
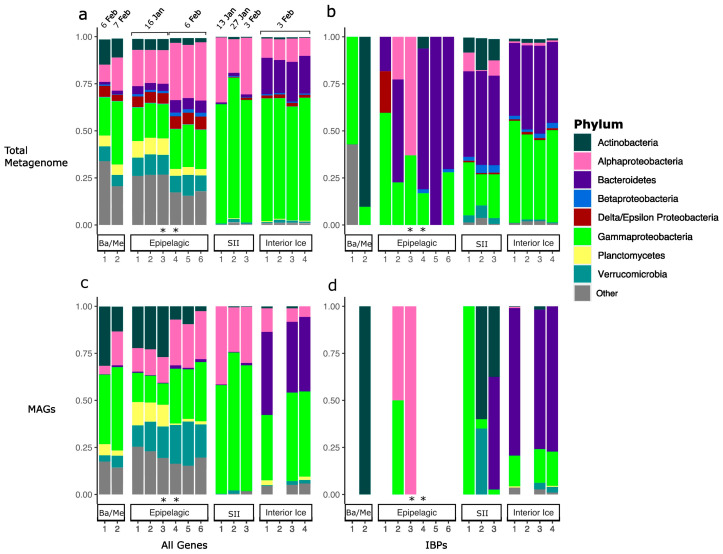
Total and ice-binding protein-encoding prokaryotic community composition and MAG distribution vary with environment type. Phylum-level composition (proportion of reads of prokaryotic assembly) of (**a**) prokaryotic whole communities and (**b**) prokaryotic taxa encoding at least one ice-binding protein (IBP). In the total assembly (**a**,**b**), whole communities were dominated by Gamma and Alpha- proteobacteria (pink and light green) across environments, with this becoming especially striking in the sea–ice interface and interior ice environments. Bacteroidetes (purple), Verrucomicrobia (teal) and Actinobacteria (dark green) were also variably dominant across environments. IBP-encoding communities were dominated by Actinobacteria in the meso/bathypelagic environment, and by Bacteriodetes and Gammaproteobacteria in all other environments. The taxonomic composition of all prokaryotic MAGs (**c**) and prokaryotic MAGs encoding at least one IBP (**d**) retrieved from these samples broadly mirrors the distribution of the total assembly communities. Ba/Me and SII refer to samples from the bathy/mesopelagic layers, and sea–ice interface (5 cm ice core bottom layer), respectively. The category ‘other’ includes all phyla with relative abundance less than 2.5%. Asterisks (*) represent samples formed by pooling. Sampling dates for all samples are indicated in panel (**a**). Note that IBP-encoding MAGs were not retrieved from each sampling location.

**Figure 3 genes-14-00363-f003:**
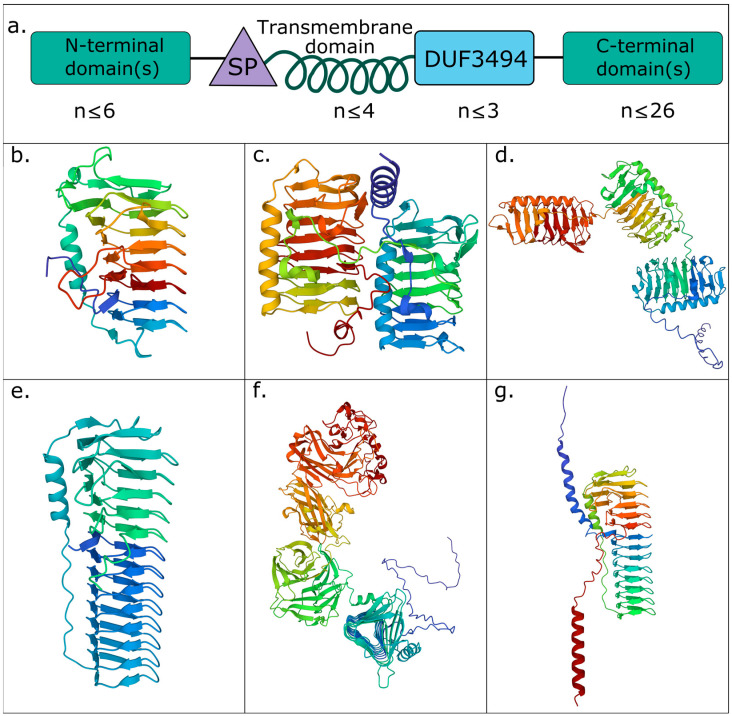
The structures and domain architectures of abundant prokaryotic ice-binding proteins reflect potentially diverse biological roles. (**a**) Concept diagram displaying the modular diversity of IBP domain architectures. IBPs minimally consist of a single DUF3494 domain (blue) but can consist of up to three DUF3494 domains. Where multiple DUF3494 domains are found, they are not necessarily found in immediate succession—other domains may be interspersed among them. Signal peptides (SP; yellow) and transmembrane domains (TMDs; green helix) are variably present, with up to four transmembrane domains being found in a protein. Additional N and C-terminal domains (pink; location defined by the position of the first DUF3494 domain) are also found, with up to 6 N-terminal domains or up to 26 C-terminal domains in a single protein (up to 28 domains in a single protein). The most abundant IBP domain architectures found in the total assembly were (**b**) single domain IBPs; (**c**) double domain IBPs; (**d**) triple domain IBPs; (**e**,**f**) single domain IBPs with an additional C-terminal DUF4842 (pfam16130); (**g**) single domain IBPs with an additional C-terminal PEP C-term motif (pfam07589). The most environmentally abundant representative of each domain architecture was selected for modelling. Note the length of the DUF3494 domain in (**e**–**g**). Proteins are coloured according to residue position, blue being the N-terminus and red being the C-terminus of the protein. Further information about domain architecture abundances is found in Table 1.

**Figure 4 genes-14-00363-f004:**
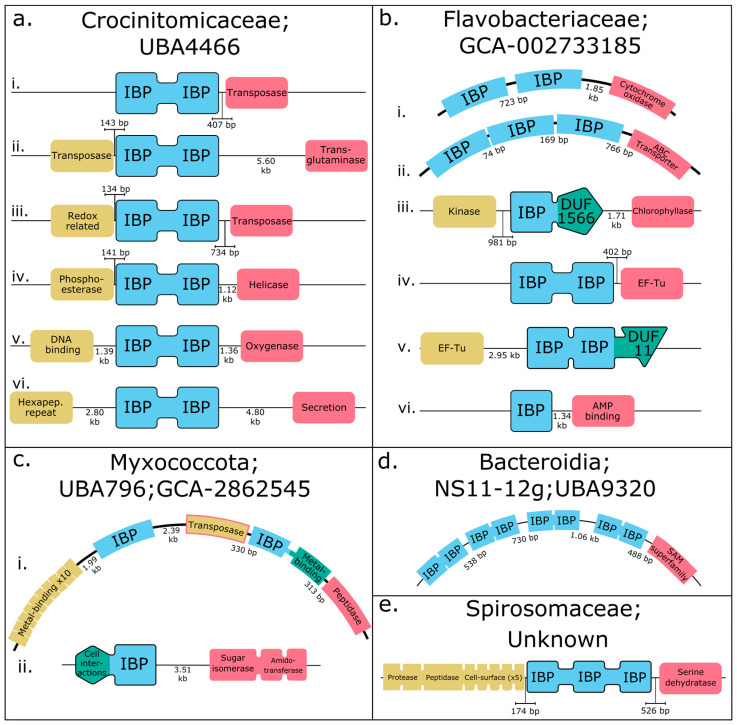
Genomic context of ice-binding proteins (IBPs) in metagenome-assembled genomes from the central Arctic Ocean. Genes upstream of IBPs are in yellow, IBPs are in blue, and genes downstream of IBPs are in pink. Double and triple domain IBPs are linked by blue connectors. Non-IBP domains within the IBP genes are in green. Each line is a single contig. Numbers between domains refer to the genomic distance between them. If an upstream or downstream domain is not shown, the IBP gene is the first or last gene in its contig. Contigs not listed in a specific order. (**a**) All six IBPs from Crocinitomicaceae (family) UBA4466 are double domain IBPs (ddIBPs). (i) ddIBP with a transposase immediately (407 bp) downstream. (ii) ddIBP with a transposase immediately (143 bp) upstream and a transglutaminase 5.60 kb downstream. (iii) ddIBP with a redox-related domain immediately (134 bp) upstream and a transposase nearby (734 bp) downstream. (iv) ddIBP with a phosphoesterase immediately (141 bp) upstream and a helicase 1.12 kb downstream. (v) ddIBP with a DNA binding domain 1.39 kb upstream and an oxygenase domain 1.36 kb downstream. (vi) ddIBP with hexapeptide repeats 2.80 kb upstream and a secretion-related domain 4.80 kb downstream. (**b**) The nine IBPs from the Flavobacteriaceae (family) GCA-002733185 are found in a variety of genomic contexts. (i) Two single domain (sd) IBPs are found adjacent to each other at the start of a contig, separated by 723 bp. A cytochrome oxidase domain is found 1.85 kb downstream of the second sdIBP. (ii) Three sdIBPs are found adjacent to each other at the start of a contig, separated by 74 and 169 bp, respectively. An ABC transporter domain is found 766 downstream. (iii) A sdIBP containing a DUF1566 domain has a kinase 981 bp upstream and a chlorophyllase 1.71 kb downstream. (iv) A ddIBP is found at the start of a contig, with an elongation factor Tu domain 402 bp downstream. (v) A ddIBP containing a DUF11 domain is found at the end of a contig, with an elongation factor Tu domain 2.95 kb upstream. (vi) A sdIBP is found at the start of a contig with an AMP-binding domain 1.34 kb downstream. (**c**) The three IBPs from Myxococcota (family); UBA796; GCA-2862545. (i) Two sdIBPs are found in the same contig, separated by a transposase. The second sdIBP also contains a metal-binding domain. A protein containing 10 metal-binding domains is found 1.99 kb upstream of the first sdIBP. A transposase is 2.39 kb downstream of the first sdIBP and 330 bp upstream of the second sdIBP. The second sdIBP is followed by a peptidase 313 bp downstream. (ii) A sdIBP with a cell-interaction related domain is found at the start of a contig, with a protein containing sugar isomerase and amidotransferase domains found 3.51 kb downstream. (**d**) All four IBPs from Bacteroidia; NS11-12g; UBA9320 are ddIBPs and are found adjacent to each other at the start of the same contig, separated by 538 bp, 730 bp and 1.06 kb, respectively, and a SAM superfamily domain is found 488 bp downstream of the fourth ddIBP. (**e**) The only IBP from Spirosomaceae (order unknown) is a triple domain IBP. A protein containing a protease domain, a peptidase domain and five cell-surface related domains is found 174 bp upstream of the IBP and a serine dehydratase domain is found 526 bp downstream. Data used to produce this figure are in Appendix A.

**Figure 5 genes-14-00363-f005:**
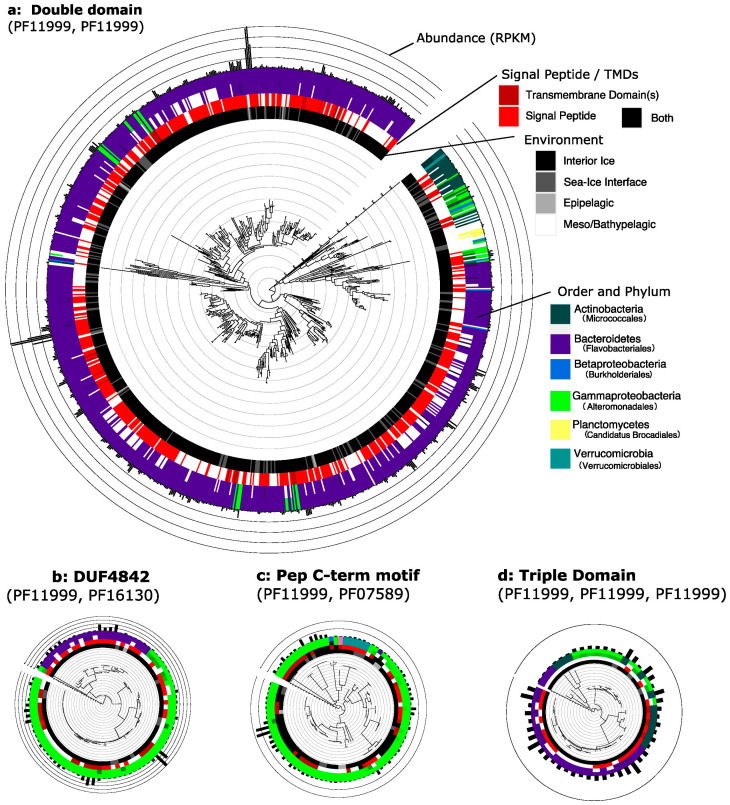
Abundant ice-binding protein domain architectures have distinct phylogenetic distributions. Gene trees of the DUF3494 domains present in the four environmentally most abundant domain architectures excluding single domain ice-binding proteins (IBPs). Trees are annotated with (from centre out) environment (black: interior ice, dark grey: sea–ice interface, light grey: epipelagic, white: meso/bathypelagic), signal peptide (SP: bright red) or transmembrane domain (TMD: dark red) presence (both: black), order-level and phylum-level classification (Bacteroidetes: purple, Gammaproteobacteria: bright green, Verrucomicrobia: teal, Actinobacteria: forest green, Betaproteobacteria: dark blue) and abundance (reads per kilobase million, demarcated in multiples of 10). Orders are coloured in shades of their parent phylum colour to show diversity within a phylum; the dominant order is shown specified in brackets in the legend and uses the same shade as the parent phylum. White gaps signify where the order was unknown. (**a**) Double domain IBPs (ddIBPs) mainly come from a single order of Bacteroidetes (Flavobacteriales), with the presence of SP and TMDs not appearing to be associated with taxonomy. (**b**) IBPs containing a C-terminal DUF4842 (pfam16130) come from Gammaproteobacteria and a single order of Bacteroidetes. TMDs are abundant only in one of the two clades of Bacteroidetes IBPs, and IBP abundance is distributed across both phyla. (**c**) IBPs containing a PEP C-term motif mainly come from Gammaproteobacteria and Verrucomicrobia, with one from each of Alphaproteobacteria, Betaproteobacteria and Bacteroidetes. The majority of these IBPs contain an SP and/or a TMD. (**d**) Triple domain IBPs (tdIBPs) come from Bacteroidetes, Gammaproteobacteria and Actinobacteria. Most tdIBPs from Bacteroidetes cluster within a single clade, which is most similar to a monophyletic clade of Actinobacteria. TdIBPs from Gammaproteobacteria are found in a clade which also contains three Actinobacteria tdIBPs. Although the majority of tdIBPs are from Bacteroidetes and are found in a monophyletic clade, the tdIBPs which are the most different from this group also contain tdIBPs from Bacteroidetes.

**Figure 6 genes-14-00363-f006:**
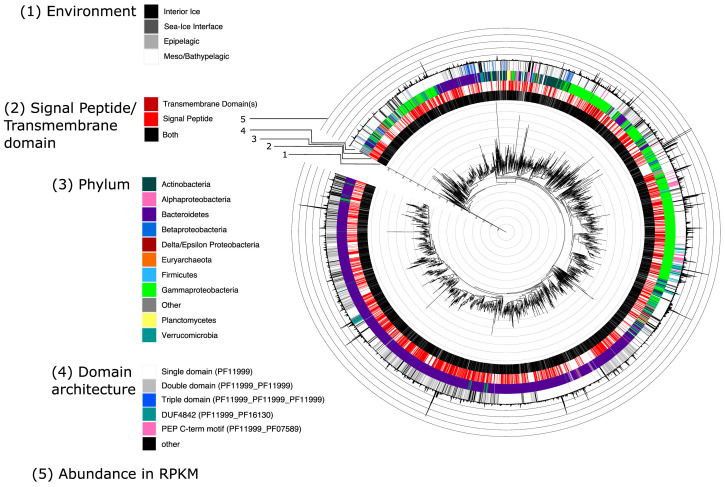
Structurally diverse prokaryotic IBPs are phylogenetically widely distributed. Trees of prokaryotic DUF3494 domains, either just found in MAGs (top), or in the total assembly. Trees are annotated with (from centre out) 1. Environment (black: interior ice, dark grey: sea–ice interface, light grey: epipelagic, white: meso/bathypelagic); 2. Signal peptide (SP: bright red) or transmembrane domain (TMD: dark red) presence (both: black); 3. Phylum-level classification (Bacteroidetes: purple, Gammaproteobacteria: bright green, Verrucomicrobia: teal, Actinobacteria: forest green, Betaproteobacteria: dark blue); 4. Domain architecture (single domain: white, double domain: grey, triple domain: blue, DUF4842: teal, PEP C-term motif: pink, other protein family: black); 5. Abundance (reads per kilobase million, demarcated in multiples of 10). Most IBPs are found in sea–ice environments. Signal peptide presence is evenly distributed across the tree. The majority of IBPs are found within Bacteroidetes or Gammaproteobacteria. Many IBPs from Bacteroidetes are more similar to each other than to IBPs from Gammaproteobacteria and other phyla. IBPs from Gammaproteobacteria cluster less closely, and are interspersed with IBPs from other phyla. Broadly, the more monophyletic, recently branched clade of the Bacteroidetes IBPs appears to be enriched with double domain IBPs and less individually abundant domain architectures, with one clustered group of DUF4842-containing IBPs also present. Conversely, the majority of IBPs with PEP C-term motifs are found within Gammaproteobacteria. Triple domain architectures are mainly found in IBPs from Gammaproteobacteria and the second-largest clade of Betaproteobacteria. Note that some of the most dissimilar IBPs come from the same phyla. IBP abundance appears to be relatively evenly distributed across phyla.

**Table 1 genes-14-00363-t001:** Abundant IBP domain architectures from the total assembly. We grouped IBPs from the total assembly by their domain architecture and summed the abundance (reads per kilobase million; RPKM) for each IBP with that architecture. We then collected their protein family names (Pfam) from the Interpro database [5]. Using information provided by Interpro, we organised each Pfam into a broader functional grouping. Note that the abundances were summed across all environments, comprising two samples from the bathy/mesopelagic zone, six samples from the epipelagic, three from the sea–ice interface and four from the interior ice.

Domain Architecture	Protein Family (Pfam)	Broader Function	Abundance (RPKM)	Abundance (%)
pfam11999	DUF3494	IBP	4983.46	61.54
pfam11999_pfam11999	DUF3494	IBP	1660.68	20.51
pfam11999_pfam16130	DUF4842 *	β-barrel Ig fold	413.49	5.11
pfam11999_pfam07589	PEP C-term motif	Sorting/Exopolysaccharides	209.68	2.59
pfam11999_pfam11999_pfam11999	DUF3494	IBP	93.11	1.15
pfam11999_pfam11999_pfam01345	DUF11	Cell wall-related	63.74	0.79
pfam11999_pfam02010	REJ domain	Membrane associated	58.44	0.72
pfam04519_pfam11999	Polymer-forming cytoskeletal	Cytoskeleton	47.68	0.59
pfam11999_pfam11999_pfam13517_pfam13517_pfam07593	FG-GAP-like repeat	Cell adhesion	44.35	0.55
	ASPIC and UnbV	Cell adhesion		
pfam11999_pfam11999_pfam13517_pfam13517_pfam13517_pfam07593	FG-GAP-like repeat	Cell adhesion	25.93	0.32
	ASPIC and UnbV	Cell adhesion		0.31
pfam11999_pfam03797	Autotransporter β-domain	Secretion	24.83	0.30
pfam13205_pfam13205_pfam11999	BIg-like domain *	Tethering	24.45	0.26
pfam11999_pfam11999_pfam02412_pfam02412_pfam02412_pfam02412_pfam02412	Thrombospondin type 3 repeat	Cell adhesion	21.36	0.25
pfam11999_pfam01391	Collagen triple helix repeat	Cell adhesion	20.53	0.24
pfam11999_pfam07603	DUF1566	Unknown	19.42	0.23
pfam11999_pfam02494_pfam02494_pfam02494	HYR domain	Cell adhesion	18.99	0.23
pfam14341_pfam11999	PilX N-terminal	Cell adhesion	18.87	0.21
pfam04862_pfam11999	DUF642	Unknown (thought to be exclusive to plants)	16.73	0.20
pfam11999_pfam04862	DUF642	Unknown (thought to be exclusive to plants)	13.87	0.17
pfam11999_pfam01345	DUF11	Cell wall-related	13.17	0.16

* These domains are considered “Ig-like”.

## Data Availability

Sequence data and metadata are available through the DOE JGI web portal under proposal ID 505419. Sample metagenome IDs are listed in Appendix A. All data used for this study are in Appendix A.

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
