# Peer review of "Genetic and Structural Diversity of Prokaryotic Ice-Binding Proteins from the Central Arctic Ocean"

_genes, 2023, doi:10.3390/genes14020363_

Round 1
Reviewer 1 Report
The article is well written and is important in the field of prokaryotic IBP.
Author Response
Many thanks for this positive review. Much appreciated.
Reviewer 2 Report
The paper presents the unique results of the investigation of protein structures of biological samples collected in the Arctic. The results obtained by the authors make an important contribution to our understanding of the diversity of ice binding proteins produced by microorganisms living on our planet in the regions which are difficult of access. The paper is worthy of publication in the journal.
Author Response
Many thanks for your positive review. This is much appreciated.
Reviewer 3 Report
Comments and Suggestions for Authors
(will be shown to authors)
Overall, an intriguing and compelling story. Could be very important in the field of polar oceanography and microbial ecology.
Most of the comments and suggested additions/changes below are for clarification for readers such as myself who may not be as familiar with these particular areas as the author(s) of the manuscript are. All mostly minor edits / semantics, as follows:
Line 36: Just after first mentioning DUF 3494, please give approximate AA size of known ones. E.g., ~200 AA domain (as described in Raymond et al. 2021, one of your existing references). Or please give range that is described in reference [5]. Also as Raymond et al. described, please also mention the full protein length of the examples in Raymond et al’s study (e.g. 253, 359 AA).
Regarding comment in lines 75-77, to help readers new to MAG studies from polar waters, it would be a good idea to mention at least two other prokaryotic MAG polar studies: specifically, one from this team, Duncan et al. 2022 (Metagenome-assembled genomes of phytoplankton microbiomes from the Arctic and Atlantic Oceans), as well as another paper that was cited by Duncan et al. 2022: Zhang W, Cao S, Ding W, Wang M, Fan S, Yang B, et al. Structure and function of the Arctic and Antarctic marine microbiota as revealed by metagenomics. Microbiome. 2020;8:1–12 Springer.
A third is Royo-Llonch et al. (2021) which is referenced later in the discussion of this manuscript (#[53]).
Lines 50-52: This mention of Ig- and Ig-like domains should be expanded a bit, as when I got to the results regarding the IBP’s containing Ig-like structures (lines 271-277), I was unsure as to what the type of structure it was (how it fit within the various examples given in Fig 3b through 3g. Please expand the section just after lines 50-52 to describe the type of structure in bacterial immunoglobulin-like repeats.
Lines 78-79 need to be reworded as (semantically) the sentence as it stands suggests that actual structure work was done. Instead, suggested: “Here, we used mgenome-informed genomics to explore the nucleotide sequence diversity and inferred amino acids conferring presumed protein structural domain diversity.” Or something to this effect, just to make it clear to the reader that all of these structures were inferred from sequence.
Similarly, in lines 85-86: “We characterized the diverse possible IBP domain architectures present based on genomic information of prokaryotic open reading frames present in metagenomes obtained from a variety of environmental samples, including in sea ice and water.”
In line 84, ending a sentence with “……sea-ice interface and interior ice would have the highest number of IBPs.” Replace this phrase to be more precise, regarding *relative abundance* not actual abundance, e.g. “….community genomes in the s-i interface and interior ice would have a higher relative abundance of metagenomes containing IBP genes, compared to relative abundances in samples of the pelagic environments.”
Methods notes:
Lines 98-99: Samples were filtered (volumes? Please provide range of volumes here, and/or include volumes in a column added to Supp Table S1. And also in lines 98-99 -- “stored on board” (at what temperature, e.g. -20oC?)
Regarding which were pooled from >1 Niskin bottle vs which were from a single bottle, this is indicated in Supp Table S1, but please also indicate which were from pooled, perhaps with a symbol above the stacked bars in Fig 2A, 2B, 2C, 2D (epi #3, epi #4).
Lines 121-122: End this sentence “…… following the low input protocol.” with an additional phrase “and epipelagic samples with the regular protocol.” So: ““…… following the low input protocol, and epipelagic samples with the regular Illumina protocol.”
Section 2.3 needs a bit more clarification for better understanding the results/discussion sections below. Specifically, when describing the 5 most abundant (relative abundance, that is, total reads per KBM), are you talking about in all samples (together pooled) or in each of the various environment categories (pelagic, ice-interface, inner-ice)? That is, specifically in line 165, the phrase “domain architectures in the environment for modeling” – in any of the environment types, or in the total dataset?
Results section notes:
Section 3.1.1, specifically line 196. “71% could be classified to order level.” Instead, “Of all (assemblies? Reads? MAG’s? or rRNA/tRNA genes?) 71% could be classified to order level.”
Line 200: Instead of (Figure 2), state (Figure 2C & 2D).
Small clarifications/additional notations needed in Figure 2: As mentioned above, put star or symbol above 5th and 6th stacked bars (corresponding to Epi-3 and Epi-4) to indicate these were from pooled water samples (multiple Niskins). Also needed in Figure 2 (described in Supp Table S1, but would be best if you could indicate with symbols, brackets, etc.) different dates corresponding to these various replicates. E.g. Epi-3 was 01/16, and Epi-4 was 02/06. This would clarify reasons for large differences seen between Epi-3 and Epi-4 for example, especially in panels B, C, and D where epipelagic samples’ community compositions are so drastically different.
Lines 239-241: Again semantics but please make it clear that you did not actually determine protein structures. For instance, first sentence in section 3.1.2, change to something like: “From genomic sequences, inferred domain architecture diversity was large. For instance, 116 unique inferred domain arcitectures were found…..”
Line 245: (Figure 3, Table 1) please change to (Figure 3C, Table 1) and similarly, in line 246: (Figure 3D, Tab 1) for triple domain.
To help the reader better understand how results shown in Table 1 relate to structures in Figure 3, please add column to Table 1 (or insets in Fig 3 panels B through G) to indicate:
(a) which of the domain architectures listed in Table 1 are relevant to the Ig-like? This is especially important given what is discussed in lines 499-503 and again 506-521. In several reads, the first instance where I get an answer to this is in line 417, where it is explicitly stated “….unknown, but which contains a beta-barrel immunoglobulin fold (Figure 5b).”
(b) relative abundance (percentages) as indicated in the text. For example, lines 271 to 277, it is stated that a large proportion of the environmental *relative* abundance (6.62%) were Ig-like. So are these percentages based on the RPKM values in the right-most column in Table 1? If so please put ( xx%) next to each of the actual RPKM values. Or if there percentages are elsewhere, please point the reader to the table showing these percentages.
Same issue with all the percentages provided in lines 278-291. But it is clearer the way you state it in lines 279-280: “IBPs with these domain arch’s represented 5.82% of the environmental *relative* abundance, comprising 4.88% of all IBPs found.”
Table 1 header (lines 295-299): “Abundant IBP domain architectures from the total assembly” (does this mean all environmental types? Indicate so in Table 1 legend. Note to the reader here “total assembly” (total environment) is comprised of 2 samples from bathy/mesopelagic, 6 samples from epipelagic, 3 from sea-ice interface and 4 from interior ice.
Discussion notes:
Lines 535-537, very important statement, followed by some excellent arguments for the need for more synteny analyses, in my opinion. Excellent!
Author Response
Line 36: Just after first mentioning DUF 3494, please give approximate AA size of known ones. E.g., ~200 AA domain (as described in Raymond et al. 2021, one of your existing references). Or please give range that is described in reference [5]. Also as Raymond et al. described, please also mention the full protein length of the examples in Raymond et al’s study (e.g. 253, 359 AA).
Response: Added an approximate size for the DUF3494 domain (line 35).
Regarding comment in lines 75-77, to help readers new to MAG studies from polar waters, it would be a good idea to mention at least two other prokaryotic MAG polar studies: specifically, one from this team, Duncan et al. 2022 (Metagenome-assembled genomes of phytoplankton microbiomes from the Arctic and Atlantic Oceans), as well as another paper that was cited by Duncan et al. 2022: Zhang W, Cao S, Ding W, Wang M, Fan S, Yang B, et al. Structure and function of the Arctic and Antarctic marine microbiota as revealed by metagenomics. Microbiome. 2020;8:1–12 Springer.
A third is Royo-Llonch et al. (2021) which is referenced later in the discussion of this manuscript (#[53]).
Response: Added suggested references and added an additional reference to Royo-Llonch et al., 2021. Bibliography has been updated accordingly.
Lines 50-52: This mention of Ig- and Ig-like domains should be expanded a bit, as when I got to the results regarding the IBP’s containing Ig-like structures (lines 271-277), I was unsure as to what the type of structure it was (how it fit within the various examples given in Fig 3b through 3g. Please expand the section just after lines 50-52 to describe the type of structure in bacterial immunoglobulin-like repeats.
Response: Added the following sentence as a description of Ig-like domains, including a reference to a review about their structure and function in bacteria:
“Ig-like domains generally consist of two antiparallel β-sheets which twist to surround a hydrophobic core, and are often associated with bacterial adhesion to a variety of substrates [13]”
Lines 78-79 need to be reworded as (semantically) the sentence as it stands suggests that actual structure work was done. Instead, suggested: “Here, we used mgenome-informed genomics to explore the nucleotide sequence diversity and inferred amino acids conferring presumed protein structural domain diversity.” Or something to this effect, just to make it clear to the reader that all of these structures were inferred from sequence.
Similarly, in lines 85-86: “We characterized the diverse possible IBP domain architectures present based on genomic information of prokaryotic open reading frames present in metagenomes obtained from a variety of environmental samples, including in sea ice and water.”
Response: Added language in the specified lines to clarify that protein structures from this study are predictions/models rather than the result of structural work.
In line 84, ending a sentence with “……sea-ice interface and interior ice would have the highest number of IBPs.” Replace this phrase to be more precise, regarding *relative abundance* not actual abundance, e.g. “….community genomes in the s-i interface and interior ice would have a higher relative abundance of metagenomes containing IBP genes, compared to relative abundances in samples of the pelagic environments.”
Response: Reworded the phrase to be more precise: “We explored total community composition as well as the composition of IBP-encoding taxa within each environment, expecting that metagenomes from the sea-ice interface and interior ice would have a higher relative abundance of IBP genes compared to those from epipelagic or meso/bathypelagic environments.”
Methods notes:
Lines 98-99: Samples were filtered (volumes? Please provide range of volumes here, and/or include volumes in a column added to Supp Table S1. And also in lines 98-99 -- “stored on board” (at what temperature, e.g. -20oC?)
Response: Added storage temperature for samples, and added a column to Supp. Table 1 with the volumes.
Regarding which were pooled from >1 Niskin bottle vs which were from a single bottle, this is indicated in Supp Table S1, but please also indicate which were from pooled, perhaps with a symbol above the stacked bars in Fig 2A, 2B, 2C, 2D (epi #3, epi #4).
Response: See amendments to Figure 2 (below).
Lines 121-122: End this sentence “…… following the low input protocol.” with an additional phrase “and epipelagic samples with the regular protocol.” So: ““…… following the low input protocol, and epipelagic samples with the regular Illumina protocol.”
Response: Amended to reference which protocol was used for each sample.
Section 2.3 needs a bit more clarification for better understanding the results/discussion sections below. Specifically, when describing the 5 most abundant (relative abundance, that is, total reads per KBM), are you talking about in all samples (together pooled) or in each of the various environment categories (pelagic, ice-interface, inner-ice)? That is, specifically in line 165, the phrase “domain architectures in the environment for modeling” – in any of the environment types, or in the total dataset?
Response: Reworded for clarity: “We selected the 5 most environmentally abundant (total reads per kilobase million; RPKM) domain architectures in the total dataset for modelling. Representative IBPs for each domain architecture were further selected on the basis of their environmental abundance.”
Results section notes:
Section 3.1.1, specifically line 196. “71% could be classified to order level.” Instead, “Of all (assemblies? Reads? MAG’s? or rRNA/tRNA genes?) 71% could be classified to order level.”
Response: Amended to clarify that this section refers to all assemblies.
Line 200: Instead of (Figure 2), state (Figure 2C & 2D).
Response: Amended accordingly.
Small clarifications/additional notations needed in Figure 2: As mentioned above, put star or symbol above 5th and 6th stacked bars (corresponding to Epi-3 and Epi-4) to indicate these were from pooled water samples (multiple Niskins). Also needed in Figure 2 (described in Supp Table S1, but would be best if you could indicate with symbols, brackets, etc.) different dates corresponding to these various replicates. E.g. Epi-3 was 01/16, and Epi-4 was 02/06. This would clarify reasons for large differences seen between Epi-3 and Epi-4 for example, especially in panels B, C, and D where epipelagic samples’ community compositions are so drastically different.
Response: Added asterisks to the figure and a description of their meaning to the caption. Added sampling dates to panel (a).
Lines 239-241: Again semantics but please make it clear that you did not actually determine protein structures. For instance, first sentence in section 3.1.2, change to something like: “From genomic sequences, inferred domain architecture diversity was large. For instance, 116 unique inferred domain arcitectures were found…..”
Response: Added a sentence to make it clear that these structures are computational predictions: “Diverse domain architectures were computationally predicted from genomic sequences of IBPs.”
Line 245: (Figure 3, Table 1) please change to (Figure 3C, Table 1) and similarly, in line 246: (Figure 3D, Tab 1) for triple domain.
Response: Amended accordingly
To help the reader better understand how results shown in Table 1 relate to structures in Figure 3, please add column to Table 1 (or insets in Fig 3 panels B through G) to indicate:
(a) which of the domain architectures listed in Table 1 are relevant to the Ig-like? This is especially important given what is discussed in lines 499-503 and again 506-521. In several reads, the first instance where I get an answer to this is in line 417, where it is explicitly stated “….unknown, but which contains a beta-barrel immunoglobulin fold (Figure 5b).”
Response: Added asterisks and a label for those abundant domain architectures containing Ig-like domains.
(b) relative abundance (percentages) as indicated in the text. For example, lines 271 to 277, it is stated that a large proportion of the environmental *relative* abundance (6.62%) were Ig-like. So are these percentages based on the RPKM values in the right-most column in Table 1? If so please put ( xx%) next to each of the actual RPKM values. Or if there percentages are elsewhere, please point the reader to the table showing these percentages.
Response: Added a column to the table with the percentages to clarify which percentages are based on RPKM. RPKM values are added next to percentages in the text to clarify which percentages are based on RPKM.
Same issue with all the percentages provided in lines 278-291. But it is clearer the way you state it in lines 279-280: “IBPs with these domain arch’s represented 5.82% of the environmental *relative* abundance, comprising 4.88% of all IBPs found.”
Response: Changed wording in previous sections to match this wording for improved clarity.
Table 1 header (lines 295-299): “Abundant IBP domain architectures from the total assembly” (does this mean all environmental types? Indicate so in Table 1 legend. Note to the reader here “total assembly” (total environment) is comprised of 2 samples from bathy/mesopelagic, 6 samples from epipelagic, 3 from sea-ice interface and 4 from interior ice.
Response: Clarification added to table 1 header: “Table 1. Abundant IBP domain architectures from the total assembly. We grouped IBPs from the total assembly by their domain architecture and summed the abundance (reads per kilobase million; RPKM) for each IBP with that architecture. We then collected their protein family name (Pfam) from the Interpro database [5]. Using information provided by Interpro, we organised each Pfam into a broader functional grouping. Note that abundances were summed across all environments, comprising of 2 samples from bathy/mesopelagic, 6 samples from epipelagic, 3 from the sea-ice interface and 4 from the interior ice.”
Discussion notes:
Lines 535-537, very important statement, followed by some excellent arguments for the need for more synteny analyses, in my opinion. Excellent!
Response: Many thanks!